# Assessment of knowledge on self-management and level of asthma control among patients attending a tertiary care center in Nepal: A cross-sectional study

Swojay Maharjan[1], Rajesh Khatri Chhetri[2], Ashok Basnet[2], Sirapa Maharjan[3], Kushal Shrestha[3]*, Shashwot Sedhain[3], Helina Singh[3], Kiran Dhonju[4]

1 Vayodha Hospital, Kathmandu, Nepal, 2 Shree Birendra Hospital, Kathmandu, Nepal, 3 Nepalese Army Institute of Health Sciences (NAIHS), Kathmandu Nepal, 4 Sukraraj Tropical and Infectious Disease Hospital, Kathmandu, Nepal

* shresthakushal2003@gmail.com

## Abstract

Asthma is a major public health challenge globally and is characterized by recurrent attacks of breathlessness and wheezing that vary in severity and frequency from person to person. With a notable burden in Nepal, where prevalence ranges from 4.2% to 8.9%, effective asthma management requires both updated pharmacological treatments and robust self-management practices, which involve monitoring symptoms and preventing exacerbations. In settings like Nepal, where asthma prevalence and resource limitations are concerns, enhancing patient education and behavioral interventions is crucial for better disease control. An analytical cross-sectional study was conducted to find the relationship between asthma control levels and asthma self-management at a tertiary care center from 25th May 2024 to 15th November 2024 in Nepal. The convenience sampling method was used. To evaluate differences in asthma self-management knowledge, a non-parametric statistical test was applied to the scores from the Asthma Self-Management Knowledge Questionnaire (ASMQ), which were categorized by independent variables. A p-value of < 0.05 was considered as statistically significant. A total of 145 patients had participated in our study. The mean (SD) transformed ASMQ score was 26.22 (13.32). Based on ACT scores, asthma control levels were classified as uncontrolled (40%), partially controlled (54.5%), and controlled (5%). The Kruskal-Wallis H test revealed a statistically significant difference in ASMQ scores across education levels (p = 0.047), occupation (p = 0.015) and subgroups of the asthma control test (p = 0.011). Post hoc analysis using the Dunn-Bonferroni test indicated a statistically significant difference in ASMQ scores between individuals with uncontrolled asthma and partially controlled asthma. The findings of our study suggest that asthma self-management knowledge is suboptimal and correlates with asthma control levels, emphasizing the

**Data availability statement:** All relevant data are within the paper and its Supporting information files.

**Funding:** The authors received no specific funding for this work.

**Competing interests:** The authors have declared that no competing interests exist.

critical importance of effective patient education and assessment in achieving optimal disease management.

---

## Introduction

Asthma is a chronic, heterogeneous inflammatory airway disorder affecting individuals across the life course and remains a leading cause of morbidity globally. It is characterized by recurrent episodes of wheeze, shortness of breath, chest tightness, and cough that fluctuate in intensity over time and are associated with variable expiratory airflow limitation measurable through spirometry [1,2]. Common triggers such as upper respiratory tract infections, allergens, pollutants, medications, exercise, and psychological stress contribute to airway inflammation and symptom exacerbations [2]. The Global Burden of Disease (GBD) study (2019) estimated that asthma affected approximately 262 million people worldwide and accounted for 21.6 million disability-adjusted life years (DALYs), highlighting its substantial public health burden [1]. In Nepal, community-based studies report a prevalence of 4.2%–8.9%, with higher estimates observed among adult populations [3]. As Nepal undergoes an epidemiological shift towards non-communicable diseases (NCDs), increasing asthma burden poses a challenge to achieving the National Multisectoral Action Plan for NCDs (2021–2025), which targets a 25% reduction in premature NCD-related mortality, including chronic respiratory diseases, by 2025 [4].

Although pharmacotherapy particularly inhaled corticosteroids and bronchodilators delivered via metered-dose or dry-powder inhalers remains central to asthma management, optimal disease control depends heavily on patients' self-management capacity [5]. Self-management encompasses adherence to treatment, correct inhaler technique, trigger avoidance, symptom monitoring, and timely medical consultation [6]. Evidence across diverse settings demonstrates that patient education targeting these domains can significantly reduce exacerbations and improve asthma control. Interventions such as structured inhaler-technique counseling, smartphone-based training tools for inhaler competency among students and allergen immunotherapy for prevention and control have shown beneficial outcomes [7–9]. The role of comorbidity-directed management is also notable. Omeprazole therapy among poorly controlled asthmatic children with gastroesophageal reflux led to improved asthma control [10]. Lifestyle-oriented strategies, including dietary intervention among obese children, have additionally been linked to reduced asthma risk and better disease control [11]. Beyond clinical outcomes, achieving asthma control is strongly associated with improved quality of life, as demonstrated in outpatient cohorts in Brazil [12].

Despite this growing body of evidence, asthma control remains suboptimal in low- and middle-income countries, where gaps in knowledge, adherence, inhaler technique, and health literacy persist. Studies consistently report that patients with inadequate understanding of self-management are more likely to have uncontrolled disease, frequent exacerbations, and poor therapeutic outcomes [5,6]. Nepal, with limited specialist access and relatively low awareness of chronic respiratory disease management, represents a context where patient-centered education and

behavior-focused interventions could make substantial impact. However, data on asthma self-management knowledge among Nepali patients remain scarce, and literature linking knowledge with disease control in this setting is minimal. Generating such evidence is crucial for guiding context-responsive educational interventions and informing public-health strategies aimed at reducing preventable morbidity.

The present study aims to assess the level of knowledge on asthma self-management among patients attending a tertiary care center in Nepal and evaluate its association with asthma control. By addressing an existing evidence gap in a resource-limited setting, this research contributes theoretically to understanding behavioral determinants of asthma outcomes in LMIC contexts and practically to informing strategies for patient education, clinical counseling, and community-level intervention design. Strengthening self-management capacity may serve as a cost-effective pathway to improve control and support national goals for reducing respiratory disease burden.

## Methods

### Study design

This is an analytical cross-sectional study conducted to find the association between levels of asthma control with knowledge on asthma self-management, and also to relate demographic profiles with level of asthma control. Assessing asthma self-management using standardized methods and connecting it to disease control makes sense in a Low-and-Middle-Income-Country (LMIC) like Nepal, where the doctor-to-patient ratio is subpar and there is a significant intellectual divide between them [13].

### Ethical approval

This study was approved by the Institutional Review Committee of the Nepalese Army Institute of Health Sciences (NAIHS). By including the written informed consent form in the questionnaire itself, all participants were made aware of the study's purpose. The consent was assumed to have been granted by the participants who filled out the form.

### Setting

The teaching hospital of NAIHS, Shree Birendra Hospital (SBH), a tertiary care centre where all patients with military ties are given free treatment and medicines, was chosen as the suitable location for this study. The patient statistics of SBH show a variation in age groups, educational level and socioeconomic status, making this study as inclusive as possible.

### Study sample

Patients with bronchial asthma for at least 6 months, aged 18 years and above, visiting the SBH Chest OPD, and who gave their written informed consent were selected for the study. A non-probability convenience sampling method was employed, whereby participants were selected based on accessibility, and data collection was conducted at times convenient to the researchers.. The patients with any major medical comorbidity, cognitive impairment, inability to communicate verbally, asthma-COPD overlap syndrome, and those who had symptoms similar to asthma but were not diagnosed were excluded from the study. All those patients who met the inclusion criteria were interviewed from 30th June 2024 to 1st January 2025.

### Study instruments

Asthma self-management behavior and asthma-related knowledge were assessed using the Asthma Self-Management Questionnaire (ASMQ) and the Asthma Control Test (ACT). Both instruments were translated into Nepali following standard translation and back-translation procedures to ensure linguistic accuracy and cultural relevance. The translated questionnaires were then reviewed by two pulmonologists and one public health expert. The Cronbach's alpha for ASMQ was 0.513 and ACT was 0.554.

The Asthma Control Test (ACT) is a validated five-item measure that evaluates the level of asthma control over the preceding four weeks. The items assess: (1) frequency of shortness of breath, (2) occurrence of nighttime asthma symptoms, (3) degree of functional limitation, (4) frequency of rescue inhaler use, and (5) the patient's self-rating of overall asthma control. Each item is scored on a 5-point Likert scale, generating a total score ranging from 5 to 25. Higher scores indicate better asthma control: scores ≥20 represent well-controlled asthma, 16–19 partially controlled asthma, and ≤15 poorly controlled asthma. The ACT has demonstrated strong reliability and validity across diverse populations.

The Asthma Self-Management Questionnaire (ASMQ) consists of 16 multiple-choice items that assess patient knowledge related to asthma triggers, preventive strategies, proper inhaler technique, and the appropriate use of rescue and controller medications, as well as peak flow monitoring. Each item has a single correct response. Total ASMQ scores are calculated by summing correct answers or converting them to a percentage scale; higher scores indicate greater asthma knowledge and more appropriate self-management practices. The ASMQ has been widely used in clinical and population-based research and exhibits good psychometric properties.

Both questionnaires were administered through face-to-face interviews to accommodate varying levels of literacy among participants. Interviewers followed standardized administration procedures, providing instructions without offering clarifications that could bias responses. The average administration time for both tools combined was approximately 10–15 minutes per participant.

In addition to these instruments, a structured interview schedule was used to collect sociodemographic and clinical data, including age, gender, occupation, educational level, family history of asthma, known allergens, cigarette smoking history, exposure to pets, dust, and dampness, and any recent history of asthma exacerbations.

## Data collection

The data was collected through a closed-question interview with multiple-choice questions over a 7-month period (2024/06/30–2025/01/30) and analyzed by the authors. A written informed consent was taken before the respondents were assigned a printed questionnaire, both of which are attached as a single file in the annex section. Interviewees were assured that the questions had no fixed correct answer and that their response would not be judged as right or wrong to limit bias in the data collection process.

## Statistical analysis

The data collected via the questionnaire was first cleaned in Microsoft Excel 2016 and imported into SPSS version 20 for the analysis. The statistical significance of the differences was ascertained from the p-value ($<0.05$). Firstly, normality was checked using Kolmogorov-Smirnov (K–S) test and the Shapiro-Wilk test. If the test's significant value was higher than 0.05, the data distribution was considered normal; if it was less than 0.05, it was considered non-normal.

Since the assumption of normality for the ASMQ score was not satisfied for all group combinations of independent variables, normality was rejected. Then we proceeded with the non-parametric tests using the median and Inter-Quartile Range (IQR) as measures of central tendency and dispersion, respectively. A Mann-Whitney U test was chosen as our study failed the assumption of independent samples t-test to determine if there were differences in ASMQ scores between two independent categorical variables like gender, occupation, family history of asthma, level of education, known allergy, history of time spent with pets, history of smoking, living in a damp and dusty environment, and exacerbations in the past 1 year. Effect sizes for each comparison were quantified using the rank biserial correlation ($r_{(rb)}$), calculated from the U statistic and group sample sizes.

Whereas, the Kruskal-Wallis H test was run to determine if there were differences in ASMQ scores between the three sub-groups of ACT and five groups of participants with different occupation categories and educational status as our study also didn't meet the requirements of the parametric One-Way ANOVA test. When statistically significant results were obtained, post-hoc pairwise comparisons were conducted using Dunn's test with Bonferroni correction. Effect sizes

for Kruskal–Wallis tests were calculated using eta-squared ($\eta^2\_H$) to quantify the proportion of variance in ASMQ scores attributable to group differences.

## Results

### General characteristics

Out of approximately 168 patients initially approached, 145 gave their consent to participate in the study. The majority of the participants in our study were female, 96 (66.2%), out of whom 65 were homemakers. Similarly, 45 (31%) had a family history of asthma, and 24 (16.6%) had an exacerbation in the last 1 year. Table 1 depicts the socio-demographic characteristics, ACT scores and level of asthma control based on the ACT and ASMQ scores of the study population. The distribution of transformed ASMQ scores among study participants is shown in Fig 1.

**Table 1.  Socio-demographic characteristics of the study population.**

| S.N | Characteristics | Subgroups | Values N (%) |
|---|---|---|---|
| 1 | Age, Mean (SD) | | 38.89 (11.60) |
| 2 | Gender | Male | 49 (33.80) |
| | | Female | 96 (66.20) |
| 3 | Occupation | Army Personnel | 34 (23.40) |
| | | Homemaker | 65 (44.80) |
| | | Small business person | 4 (2.80) |
| | | Farmer | 9 (6.20) |
| | | Others | 33 (22.80) |
| 4 | Family history of asthma | Present | 45 (31.00) |
| | | Absent | 100 (69.00) |
| 5 | Level of education | Illiterate | 9 (6.20) |
| | | Literate | 45 (31.00) |
| | | Secondary | 44 (30.30) |
| | | Higher-secondary | 37 (25.50) |
| | | University | 10 (6.90) |
| 6 | Known allergy | Present | 76 (52.40) |
| | | Absent | 69 (47.60) |
| 7 | Time spent with pets | Present | 33 (22.80) |
| | | Absent | 112 (77.20) |
| 8 | Smoking History | Present | 10 (6.90) |
| | | Absent | 135 (93.10) |
| 9 | Damp and dusty room | Present | 14 (9.70) |
| | | Absent | 131 (90.30) |
| 10 | Exacerbations in the past 1 year | Present | 24 (16.60) |
| | | Absent | 121 (83.40) |
| 11 | ACT score Mean (SD) | | 20.03 (3.12) |
| 12 | Level of Asthma control based on ACT | Controlled | 8 (5.50) |
| | | Partially controlled | 79 (54.50) |
| | | Uncontrolled | 58 (40.00) |
| 13 | ASMQ score Mean (SD) | | 26.22 (13.32) |

SD: Standard deviation, N: Frequency, %: Percentage.

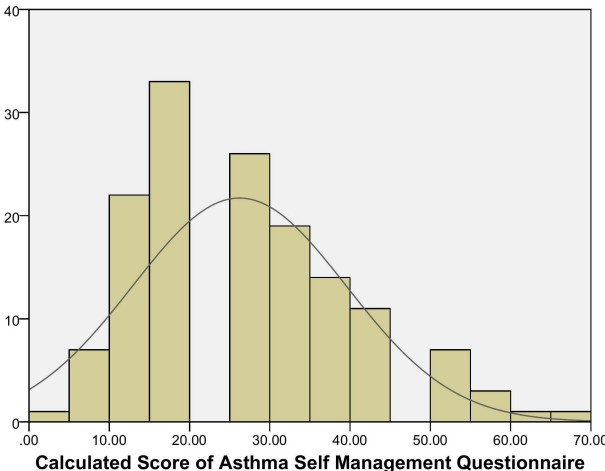

**Fig 1. Distribution of the ASMQ transformed score: possible range 0–100, with higher scores indicating more knowledge of asthma self-management.**

Fig 2 demonstrates that item 7 has the largest percentage of correct responses (75.86%) (When using your inhaler, you should inhale slowly) and the lowest percentage of correct responses (4.83%) is item 13 (Asthma can be cured by, there is no known cure for asthma). Similarly, the highest percentage of incorrect answer is for the item 13. Whereas, the questions with the highest percentage of unknown or "Don't know" answer is item 10 (Taking more rescue medicines than prescribed may mean you need more maintenance medicine). Similarly, Table 2 depicts the characteristics of incorrect responses in the ASMQ questionnaire.

Using the Mann-Whitney U test, the distributions of the ASMQ scores for males and females were assessed by visual inspection which were similar. Median ASMQ score for males (25) and females (25) was not statistically significantly different, $U = 1,895$, $z = -1.926$, $p = 0.054$, using an exact sampling distribution for $U$. The corresponding rank biserial correlation indicated a small effect size ($r_{(rb)} = 0.194$), with females showing slightly lower score variability. Similarly, the box-plots distributions were similar for family history and allergen history, however their respective median values were not statistically significantly different. The effect sizes of comparison between the ASMQ scores of the remaining dichotomous variables ranged from –0.154 to 0.162, indicating negligible to small effects.

Using the Kruskal-Wallis H test, the differences in ASMQ score between five groups of participants with different occupation and educational status were assessed. Distributions of ASMQ scores were not similar for all groups in both categories, as assessed by visual inspection of a boxplot. However, the distributions of ASMQ scores were statistically significantly different between educational status, $\chi^2(4) = 9.651$, $p = .047$ and occupation $\chi^2(4) = 12.392$, $p = 0.015$.

Subsequently, pairwise comparisons were performed using Dunn's (1964) procedure with a Bonferroni correction for multiple comparisons. Adjusted p values are presented. This post hoc analysis revealed statistically significant differences in median ASMQ scores between Farmer-Others ($p = 0.048$) and Farmer-Army Personnel ($p = 0.020$) under the Occupation category and between the Literate-University ($p = 0.033$) under the Level of Education category. The associated effect sizes were η²_H = 0.040 for education and η²_H = 0.060 for occupation, indicating small-to-moderate effects.

Further relationship between ASMQ scores and independent categorical variables are shown by Table 3.

A Kruskal-Wallis H test was run to determine if there were differences in ASMQ score between three sub-groups of ACT scores. Distributions of ASMQ scores were similar for all groups, as assessed by visual inspection of a boxplot. The distributions of ASMQ scores were statistically significantly different between groups, $\chi^2[2] = 8.972$, $p = 0.011$ with an effect size of η²_H = 0.049, suggesting a small-to-moderate association between asthma control level and self-management

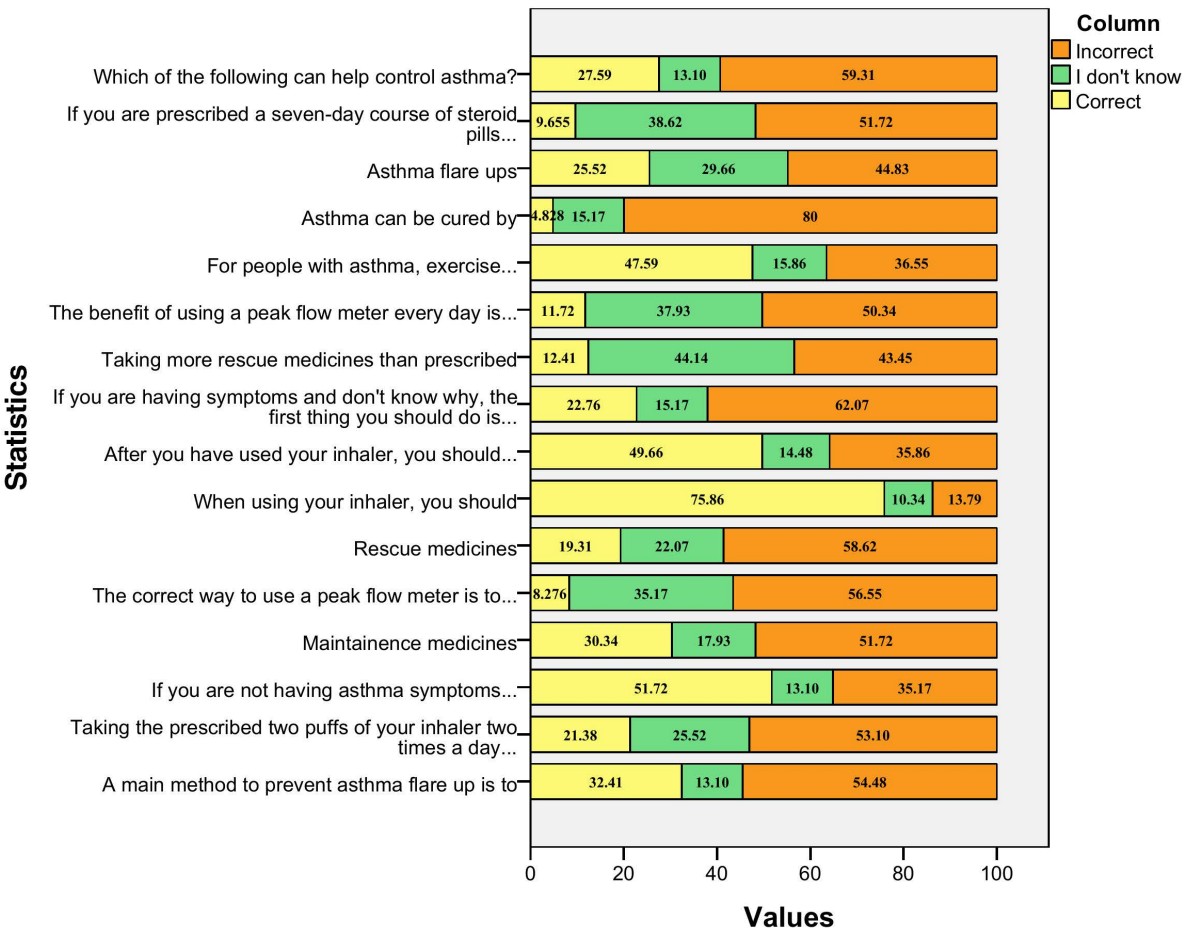

**Fig 2. ASMQ frequency distribution among the study participants with asthma.**

knowledge. The post hoc analysis revealed statistically significant differences in median ASMQ scores between the uncontrolled (22.3284) and partially controlled (28.9004) ($p = 0.010$). Adjusted $p$-value is presented in Table 4.

## Discussion

The design and implementation of this study has allowed several inferences on the knowledge of self-management of asthma in relation to various socio-demographic factors and the extent of control of the disease. Despite different studies making modifications to the ASMQ by excluding questions about peak flow meter because of the tendency of high scores among its users, our study has adopted the original form by Mancuso et al.[14]

The questionnaire assessed several components of asthma self-management. Only 32.41% participants correctly identified that flu vaccination prevents asthma flare-ups, while 54.48% responded incorrectly and 13.1% were unaware. Similarly, 44.83% had misconceptions regarding how flare-ups occur and 29.66% did not know the mechanism. Concerning dose regimen, 21.38% participants were aware that *two puffs twice daily* is not equivalent to other combinations; however, 53.10% considered other combinations the same and 25.52% did not know the correct regimen. When asymptomatic, 51.72% acknowledged the need to avoid triggers, whereas 35.17% responded incorrectly and 13.1% were unaware. Only 30.34% recognized the preventive purpose of maintenance medicines, with 51.72% responding incorrectly and 17.93%

PLOS Global Public Health | https://doi.org/10.1371/journal.pgph.0006563   May 29, 2026

**Table 2.  Characteristics of incorrect responses to the ASMQ questions (N = 145 respondents) ASMQ.**

**Misunderstandings about maintenance and rescue medications, and their proper use.**

| S.N | ASMQ Items | Incorrect responses | N (%) |
|---|---|---|---|
| 1 | A main method to prevent asthma flare up is to… | Take medicines before meals | 35 (24.1) |
| | | Take steroids in pill form | 11 (7.6) |
| | | Go to the emergency room at the first sign of symptoms | 33 (22.8) |
| 2 | Maintainence medicines… | Don't need to be taken every day | 16 (11.0) |
| | | Make you breathe better right after you take them | 54 (37.2) |
| | | Can only be taken in pill form | 5 (3.4) |
| | | Start exhaling and then put the mouth- piece in your mouth | 28 (19.3) |
| | | Put the mouthpiece in your mouth and then inhale and exhale | 19 (13.1) |
| 3 | Rescue medicines… | Help prevent future flare-ups | 35 (24.1) |
| | | Have no side effects | 37 (25.5) |
| | | Do not cause you to become tolerant to medicine | 13 (9.0) |
| 4 | Taking more rescue medicines than prescribed… | Is really not harmful | 40 (27.6) |
| | | Is a good way to manage symptoms caused by exercise | 16 (11.0) |
| | | May mean you can take less maintenance medicine | 7 (4.8) |
| **Misconceptions about the correct techniques for using inhalers and peak flow meters and its benefits.** | | | |
| 5 | The correct way to use a peak flow meter is to... | Take a deep breath and then blow into the mouthpiece slowly | 35 (24.1) |
| | | Start exhaling and then put the mouth- piece in your mouth | 28 (19.3) |
| | | Put the mouthpiece in your mouth and then inhale and exhale | 19 (13.1) |
| 6 | When using your inhaler, you should… | Take shallow breaths | 2 (1.4) |
| | | Inhale quickly | 15 (10.3) |
| | | Press your inhaler several times while you are inhaling | 3 (2.1) |
| 7 | After you have used your inhaler, you should... | Take the second puff as soon as possible after the first puff | 16 (11.0) |
| | | Keep taking puffs until you feel better | 23 (15.9) |
| | | Wash the inhaler in a tub of water | 13 (9.0) |
| 8 | The benefit of using a peak flow meter every day is... | It can tell you when you can decrease your medicines | 28 (19.3) |
| | | You can see how well you can inhale | 33 (22.8) |
| | | You can have a way to compare yourself to other people with asthma | 12 (8.3) |
| **Misunderstandings regarding symptom management, flare-ups, the implications of not experiencing symptoms and others.** | | | |
| 9 | Taking the prescribed two puffs of your inhaler two times a day... | Is the same as taking one puff four times a day | 43 (29.7) |
| | | Is the same as taking four puffs once a day | 25 (17.2) |
| | | Can be arranged in any way as long as you take a total of four puffs a day | 9 (6.2) |
| 10 | If you are not having asthma symptoms... | Your lungs are not sensitive to irritants | 13 (9.0) |
| | | It is OK to skip some doses of medicine | 26 (17.9) |
| | | You are probably cured of asthma | 12 (8.3) |
| 11 | If you are having symptoms and don't know why, the first thing you should do is... | Take some doses of steroid medicine | 8 (5.5) |
| | | Call your doctor | 65 (44.8) |
| | | Count how fast you are breathing | 17 (11.7) |

*(Continued)*

**Table 2.** (Continued)

**Misunderstandings about maintenance and rescue medications, and their proper use.**

| S.N | ASMQ Items | Incorrect responses | N (%) |
|---|---|---|---|
| 12 | For people with asthma, exercise... | Is something that should not be done regularly | 7 (4.8) |
| | | Is only good if done for at least 30 minutes at a time | 33 (22.8) |
| | | Can trigger symptoms because the lungs are not taking in enough oxygen | 13 (9.0) |
| 13 | Asthma can be cured by… | Taking daily medicine | 54 (37.2) |
| | | Avoiding triggers, such as dust and cigarette smoke | 51 (35.2) |
| | | Using a peak flow meter | 11 (7.6) |
| 14 | Asthma flare ups… | Usually occur suddenly without warning | 21 (14.5) |
| | | Cannot be triggered by strong emotions | 17 (11.7) |
| | | Always cause wheezing | 27 (18.6) |
| 15 | If you are prescribed a seven-day course of steroid pills... | You don't have to avoid triggers while you are taking the pills | 10 (6.9) |
| | | Your symptoms can't get worse while you are taking the pills | 41 (28.3) |
| | | You don't need to use your peak flow meter while you are taking the pills | 24 (16.6) |
| 16 | Which of the following can help control asthma? | Reducing stress levels | 23 (15.9) |
| | | Drinking plenty of water to stay hydrated | 46 (31.7) |
| | | Avoiding foods with sulfites, such as dried fruits and wine | 17 (11.7) |

N: Frequency, %: Percentage.

uncertain. Similar deficiencies were observed in understanding the mechanisms of asthma exacerbations, with a considerable proportion of participants either holding misconceptions or lacking knowledge altogether. Comparable trends have been reported in earlier studies, where inadequate disease understanding was associated with poor asthma control and increased exacerbation rates [13,15].

Knowledge related to peak flow meter use was particularly poor. Only 8.276% knew the correct technique and 11.72% recognized its benefit in identifying early lung function decline. In contrast, knowledge regarding inhaler technique was relatively better: 75.86% identified slow inhalation as correct and 49.66% knew breath-holding after actuation. Regarding rescue medicine usage, 19.31% knew it should not exceed 3–4 times/day and 12.41% knew that excessive rescue medicine use increases the requirement for maintenance therapy. Our findings of peak flow meter, inhaler technique, reliever and preventer medicines are consistent with the findings from a systematic review done by Alyas et al [15].

Our study showed a significant burden of poorly managed asthma, as just 8 patients (5.5%) had well-controlled asthma, while the majority had either uncontrolled asthma (40.0%) or moderately controlled asthma (54.5%). These results are in line with data from the Asthma Insights and Reality in Asia-Pacific Study, which showed that asthma control in the Asia-Pacific area is still not at its best, with many patients having chronic symptoms and inadequate treatment [16]. Similarly, data from the Asia-Pacific Asthma Insights and Management (AP-AIM) survey reported that nearly 90% of patients had partly controlled or uncontrolled asthma, with only a small minority achieving optimal control [17].

In this study, education level (p = 0.047) and occupation (p = 0.015) had significant association with ASMQ scores (p < 0.05). This aligns with Nguyen et al in Vietnam, where college-educated patients had higher ASMQ scores [18], and with Khor et al in Singapore, which also demonstrated a positive association between education and asthma knowledge

**Table 3. Relationship between ASMQ score and independent categorical variables.**

| Characteristics | | ASMQ score Median (Interquartile range) | P value | Rank biserial correlation ($r_trb_i$) and Kruskal-Wallis Effect Sizes ($\eta^2\_H$) Value |
|---|---|---|---|---|
| Gender | Male | 25.00 (25.00) | 0.054 | 0.194 |
| | Female | 25.00 (16.04) | | |
| Occupation | Army Personnel | 25.00 (25.00) | 0.015 | 0.060 |
| | Homemaker | 25.00 (17.27) | | |
| | Small business person | 15.21 (21.41) | | |
| | Farmer | 6.25 (15.63) | | |
| | Others | 25.00 (18.75) | | |
| Family history of asthma | Present | 25.00 (15.63) | 0.115 | 0.162 |
| | Absent | 25.00 (22.29) | | |
| Level of education | Illiterate | 18.75 (25.00) | 0.047 | 0.040 |
| | Literate | 18.75 (17.27) | | |
| | Secondary | 25.00 (25.00) | | |
| | Higher-secondary | 25.00 (15.63) | | |
| | University | 34.38 (14.06) | | |
| Known allergy | Present | 25.00 (19.97) | 0.883 | 0.014 |
| | Absent | 25.00 (22.29) | | |
| Time spent with pets | Present | 25.00 (17.40) | 0.794 | 0.030 |
| | Absent | 25.00 (22.29) | | |
| Smoking History | Present | 21.88 (26.56) | 0.640 | 0.088 |
| | Absent | 25.00 (22.29) | | |
| Damp and dusty room | Present | 18.75 (17.60) | 0.601 | -0.085 |
| | Absent | 25.00 (22.29) | | |
| Exacerbations in the past 1 year | Present | 18.75 (18.75) | 0.231 | -0.154 |
| | Absent | 25.00 (22.29) | | |

**Table 4. Relationship between ASMQ and levels of asthma control based on the ACT category.**

| ACT sub groups | Mean | Number | Std. Deviation | Grouped Median | Interquartile range | P value | Kruskal-Wallis Effect size ($\eta^2_h$) Value |
|---|---|---|---|---|---|---|---|
| Controlled | 28.0213 | 8 | 11.08958 | 26.5625 | 22.97 | 0.011* | 0.049 |
| Partially controlled | 28.9004 | 79 | 13.50686 | 27.3148 | 18.75 | | |
| Uncontrolled | 22.3284 | 58 | 12.56868 | 19.0789 | 18.75 | | |
| Total | 26.2231 | 145 | 13.32476 | 24.1071 | | | |

* The p value represents the statistical test result comparing the ASMQ score across the three sub groups of ACT.

[19]. Low educational attainment may hinder understanding of health messages and patient counselling. Studies from Pakistan and Egypt showed that structured educational interventions improve ASMQ scores, suggesting the benefit of patient education where clinical teaching time is constrained [20,21]. However, the present study did not measure the extent of prior patient education.

ASMQ score also differed among ACT subgroups (p = 0.010), indicating better asthma control with higher self-management knowledge. While Nguyen et al reported a similar trend of high ACT corresponding with high ASMQ [18], Khor et al found no significant association between asthma knowledge and asthma control [19], suggesting that cultural, healthcare access, or medication adherence differences may affect this relationship.

This study highlights the role of self-management knowledge in achieving better asthma control. However, certain methodological limitations need to be acknowledged. The use of self-reported data may introduce reporting bias, and the convenience sampling technique limits generalizability of the findings. Future research could incorporate objective measures of disease control, such as pulmonary function testing, adopt probability sampling to minimize selection bias, and include an educational intervention to more accurately evaluate the hypothesis.

Evidence from other settings also supports the potential value of educational interventions. A quasi-experimental study in Nigeria reported that pharmacist-led interventions significantly improved asthma control through enhanced medication adherence and appropriate use of inhaler [22]. Similarly, a randomized clinical trial in Nigeria demonstrated better asthma control and improved adherence in patients who received educational intervention compared to who did not [23]. In the national context, Bhattarai et al. conducted an interventional study in a resource limited center from Nepal and observed improved compliance following an educational intervention, although a significant improvement in disease control was not reported [24]. Collectively, these findings suggest that while knowledge alone may not guarantee improved outcomes, structured educational interventions hold promise and warrant further investigation in larger, rigorously designed studies.

There are some limitations of this study. First, the use of a non-probability convenience sampling method from a single tertiary care center may limit the generalizability of the results to the broader population. Second, the data were collected through interviewer-administered questionnaires, which may introduce reporting and social desirability bias. Third, the internal consistency of the translated instruments in this study was relatively low, with Cronbach's alpha values of 0.513 for the ASMQ and 0.554 for the ACT. This may reflect cultural and linguistic differences, variation in patient understanding, or the multidimensional nature of the constructs being measured, and suggests potential measurement error that could affect the precision of the observed associations. Additionally, potential confounders such as prior exposure to asthma education, medication adherence, and objective measures of disease control (e.g., spirometry) were not evaluated. Finally, while several socio-demographic and environmental variables were included, their relationship with asthma control was not explored, which may have provided further insights into determinants of disease outcomes.

## Conclusion

This study demonstrates that asthma self-management knowledge among patients in a tertiary care setting in Nepal is suboptimal, with a high proportion of patients exhibiting partially controlled and uncontrolled disease. The observed association between higher self-management knowledge and better asthma control highlights the clinical importance of patient education as a core component of asthma care.

Strengthening self-management programs through structured educational interventions and objective assessment tools may therefore play an important role in improving long-term clinical outcomes, particularly in resource-limited settings.

## Supporting information

**S1 File. Questionnaire.** a. The first page consists of the informed consent form. b. Second page consists of the socio demographic characteristics. c. Third page consists of the 5 items of the Asthma Control Test. d. The remaining ones consists of the 16 items of Asthma Self-Management Questionnaire.
(DOCX)

**S1 Data. Supplementary SPSS data file.**
(XLSX)

## Author contributions

**Conceptualization:** Swojay Maharjan, Ashok Basnet.

**Data curation:** Rajesh Khatri Chhetri, Ashok Basnet, Sirapa Maharjan, Kushal Shrestha, Shashwot Sedhain, Helina Singh, Kiran Dhonju.

**Formal analysis:** Swojay Maharjan, Kushal Shrestha.

**Investigation:** Rajesh Khatri Chhetri, Ashok Basnet, Sirapa Maharjan, Kushal Shrestha.

**Methodology:** Swojay Maharjan.

**Project administration:** Rajesh Khatri Chhetri.

**Resources:** Swojay Maharjan, Rajesh Khatri Chhetri, Sirapa Maharjan, Kushal Shrestha.

**Supervision:** Rajesh Khatri Chhetri, Ashok Basnet.

**Writing – original draft:** Swojay Maharjan, Sirapa Maharjan, Kushal Shrestha, Shashwot Sedhain, Helina Singh, Kiran Dhonju.

**Writing – review & editing:** Swojay Maharjan, Rajesh Khatri Chhetri, Ashok Basnet, Sirapa Maharjan, Kushal Shrestha, Shashwot Sedhain, Helina Singh, Kiran Dhonju.

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
