## [Decision Letter · Decision Letter 0]

31 Oct 2025

PGPH-D-25-01721

Assessment of Knowledge on Self-Management and Level of Asthma Control among Patients Attending a Tertiary Care Center in Nepal: A Cross-Sectional Study

Dear Dr. Shrestha,

Thank you for submitting your manuscript to PLOS Global Public Health. After careful consideration, we feel that it has merit but does not fully meet PLOS Global Public Health’s publication criteria as it currently stands. Therefore, we invite you to submit a revised version of the manuscript that addresses the points raised during the review process.

The manuscript has been evaluated by two reviewers, and their comments are available below.

The reviewers have raised a number of concerns. They request additional information on methodological aspects of the study and that the study acknowledges the limitations of self-reported data and selection bias. Could you please carefully revise the manuscript to address all comments raised?

We note a reviewer requested citations. Please note that these request are optional.

We look forward to receiving your revised manuscript.

Kind regards,

Katrien G. Janin, PhD

Staff Editor

Journal Requirements:

1.In the online submission form, you indicated that The data that support the findings of the study are available from the corresponding author upon reasonable request.

3. Uploaded as supplementary information.

Reviewers' comments:

Reviewer's Responses to Questions

**Comments to the Author**

1. Does this manuscript meet PLOS Global Public Health’s publication criteria? Is the manuscript technically sound, and do the data support the conclusions? The manuscript must describe methodologically and ethically rigorous research with conclusions that are appropriately drawn based on the data presented.

Reviewer #1: Partly

Reviewer #2: Yes

2. Has the statistical analysis been performed appropriately and rigorously?

Reviewer #1: Yes

Reviewer #2: Yes

3. Have the authors made all data underlying the findings in their manuscript fully available (please refer to the Data Availability Statement at the start of the manuscript PDF file)?

Reviewer #1: No

Reviewer #2: Yes

4. Is the manuscript presented in an intelligible fashion and written in standard English?

Reviewer #1: No

Reviewer #2: Yes

5. Review Comments to the Author

Reviewer #1: Thank you for your submission. The manuscript addresses an important gap in asthma management in Nepal and presents promising data. However, significant revisions are needed to ensure methodological transparency, clarity, and compliance with PLOS editorial standards. I recommend major revision and encourage resubmission following the above guidance.

A. Introduction:

First of all, in the introduction part, it is necessary to emphasize the different aspects of the study from other studies in the literature. However, the theoretical and practical contributions of the research are not emphasized enough in the conclusion part. In addition, the relevant literature has not been adequately covered. I recommend that the following studies be included in the research.

• Batard T, Taillé C, Guilleminault L, et al. Allergen Immunotherapy for the Prevention and Treatment of Asthma. Clin Exp Allergy. 2025;55(2):111-141. doi:10.1111/cea.14575

• Sikorska-Szaflik H, Połomska J, Sozańska B. The Impact of Dietary Intervention in Obese Children on Asthma Prevention and Control. Nutrients. 2022;14(20):4322. Published 2022 Oct 15. doi:10.3390/nu14204322

• Ghozali MT, Mutiara TA. Promoting knowledge of metered dose inhaler (MDI) usage among pharmacy professional students through a mobile app. J Asthma. 2024;61(8):835-846. doi:10.1080/02770903.2024.2306622

• Yagoubi A, Laid Y, Smati L, Nafissa Benhalla K, Benhassine F. Does omeprazole improve asthma-control in poorly-controlled asthmatic children with gastro-esophageal reflux. J Asthma. 2022;59(6):1169-1176. doi:10.1080/02770903.2021.1917606

• Jafari M, Sobhani M, Eftekhari K, Malekiantaghi A, Gharagozlou M, Shafiei A. The Effect of Oral Montelukast in Controlling Asthma Attacks in Children: A Randomized Double-blind Placebo Control Study. Iran J Allergy Asthma Immunol. 2023;22(5):413-419. Published 2023 Oct 29. doi:10.18502/ijaai.v22i5.13990

• Lourenco CB, Martins F, Fiss E, Grumach AS. Impact of asthma control on quality of life in an outpatient setting in Brazil. J Asthma. 2023;60(4):794-801. doi:10.1080/02770903.2022.2097092

B. Scientific and Methodological Rigor

• The research question is relevant, and the study design is broadly appropriate. However, the use of convenience sampling introduces bias and limits generalizability. Please discuss this limitation more explicitly in the Discussion.

• While the cross-sectional design is acceptable, the causal language in parts of the discussion and conclusion is overstated. The findings show association—not causation—and should be interpreted accordingly.

C. Statistical Analysis

• The use of non-parametric tests (Mann-Whitney U and Kruskal-Wallis) is appropriate given the non-normal distribution.

• However, a more detailed justification for the selection of these tests, including the rationale for rejecting normality (e.g., test statistics), would improve transparency.

• In Table 3, clarify if multiple comparisons were adjusted for when reporting multiple p-values.

• Provide effect sizes (e.g., median differences, rank biserial correlation) where possible to contextualize statistical significance.

D. Data Availability

• As per PLOS policy, data underlying all findings must be publicly available. Currently, your Data Availability Statement notes that data are available “upon reasonable request.” This does not meet PLOS standards.

• Please provide de-identified raw data (including ASMQ and ACT item-level scores) as supplementary files or deposit them in an open-access repository (e.g., Dryad, Zenodo).

E. Language and Clarity

• The manuscript contains several grammatical errors, redundancies, and non-standard phrasing that affect readability.

• Examples include:

o “Convenient method of sampling” → should be “convenience sampling method.”

o “At researchers’ appropriate time” → unclear; please rephrase or remove.

o “Median ASMQ score for males (25) and females (25)” → should include IQRs for clarity.

• A thorough language and grammar revision is strongly recommended before resubmission.

F. Presentation of Results

• Tables and figures are helpful, but some formatting is inconsistent. For example, Table 2 could be better structured for readability (group similar misconceptions together).

• Figures (esp. Fig 2) should include legends and clearer labeling for interpretation.

G. Discussion and Literature Contextualization

• The discussion fairly addresses previous studies but could be improved by:

o Emphasizing context-specific implications for LMICs like Nepal.

o Clearly acknowledging the limitations of self-reported data and selection bias.

o Integrating additional recent literature to enrich the interpretation of findings.

Wish you all the best!

Reviewer #2: The manuscript reads well and highlights an interesting correlation between poor knowledge and poor control.

Comments on specific sections:

P6

'Refusal to participate' should be removed from the Study Sample section

P7

A little too much information in the Statistical Analysis section eg detail on checking whether the data were distributed normally.

In the study instruments section, please provide some more detail about the questionnaires (eg higher ASMQ score implies more knowledge) and their administration.

P13

Table 4: It is unclear as to which comparison the p of 0.011 refers to

P14

The current content of the Discussion could be trimmed a little and it would be good to hear more about how one might go about improving knowledge and whether this might affect control.

6. PLOS authors have the option to publish the peer review history of their article (what does this mean?). If published, this will include your full peer review and any attached files.

**Do you want your identity to be public for this peer review?** For information about this choice, including consent withdrawal, please see our Privacy Policy.

Reviewer #1: No

Reviewer #2: No

Figure Resubmissions:

---

## [Decision Letter · Decision Letter 1]

19 Mar 2026

PGPH-D-25-01721R1

Assessment of Knowledge on Self-Management and Level of Asthma Control among Patients Attending a Tertiary Care Center in Nepal: A Cross-Sectional Study

Dear Dr. Shrestha,

Thank you for submitting your manuscript to PLOS Global Public Health. After careful consideration, we feel that it has merit but does not fully meet PLOS Global Public Health’s publication criteria as it currently stands. Therefore, we invite you to submit a revised version of the manuscript that addresses the points raised during the review process.

The manuscript has been assessed by a third reviewer, who requests further clarification. Their comments are available below, please address all of them carefully.

We look forward to receiving your revised manuscript.

Kind regards,

Alejandro Torrado Pacheco, PhD

Staff Editor

**Journal Requirements:**

**Additional Editor Comments (if provided):**

Reviewers' comments:

Reviewer's Responses to Questions

**Comments to the Author**

1. If the authors have adequately addressed your comments raised in a previous round of review and you feel that this manuscript is now acceptable for publication, you may indicate that here to bypass the “Comments to the Author” section, enter your conflict of interest statement in the “Confidential to Editor” section, and submit your "Accept" recommendation.

Reviewer #3: All comments have been addressed

2. Does this manuscript meet PLOS Global Public Health’s publication criteria? Is the manuscript technically sound, and do the data support the conclusions? The manuscript must describe methodologically and ethically rigorous research with conclusions that are appropriately drawn based on the data presented.

Reviewer #3: Yes

3. Has the statistical analysis been performed appropriately and rigorously?

Reviewer #3: Yes

4. Have the authors made all data underlying the findings in their manuscript fully available (please refer to the Data Availability Statement at the start of the manuscript PDF file)?

Reviewer #3: Yes

5. Is the manuscript presented in an intelligible fashion and written in standard English?

Reviewer #3: Yes

6. Review Comments to the Author

**Reviewer #3:** Comments

Section comment

Methods

Study Design “Assessing asthma self-management using standardized methods and connecting it to disease control makes sense in a Low-and-Middle-Income-Country (LMIC) like Nepal where the doctor-to-patient ratio is subpar and there is a significant intellectual divide between them.”

Provide reference of this line

Study Sample “A nonprobability convenient sampling method was used – where the convenience to the researchers was in the process of data collection, which was done at researchers’ appropriate time.”

Make this sentence grammatically correct

Study Instrument Author written both ACQ and ASMQ were translated but did not report about validity and reliability check of translated instrument.

Results Some sociodemographic characteristics like “time spent with pets” and “damp and dusty room” does not make much sense for checking association with asthma self-management knowledge Score (ASMQ), but this could make more sense if author check these variable association with asthma control test score as pet and damp, dusty room can affect Asthma control status. (But since author did not check association between sociodemographic and asthma control score, this is just suggestion)

Any particular reason for not checking association between age and ASMQ.

Discussion In the discussion the 2nd and 3rd paragraph is simply the replication of results. Author did not related these findings with existing literature, so it is suggested for author to related it to existing literature.

Author could also include asthma control status and compare it with existing literature i.e. how many are having partially controlled, uncontrolled asthma etc.

Conclusion “We found that patients’ attitudes and practices were closely linked to their knowledge of asthma self-management”.

The study measure Asthma control and self-management knowledge. So how author concluded regarding attitude?

7. PLOS authors have the option to publish the peer review history of their article (what does this mean?). If published, this will include your full peer review and any attached files.

**Do you want your identity to be public for this peer review?** For information about this choice, including consent withdrawal, please see our Privacy Policy.

Reviewer #3: **Yes:** Rabia Hussain

Figure Resubmissions:

---

## [Decision Letter · Decision Letter 2]

12 May 2026

Assessment of Knowledge on Self-Management and Level of Asthma Control among Patients Attending a Tertiary Care Center in Nepal: A Cross-Sectional Study

PGPH-D-25-01721R2

Dear Mr. Shrestha,

We are pleased to inform you that your manuscript 'Assessment of Knowledge on Self-Management and Level of Asthma Control among Patients Attending a Tertiary Care Center in Nepal: A Cross-Sectional Study' has been provisionally accepted for publication in PLOS Global Public Health.

Best regards,

Julia Robinson

Executive Editor

Reviewer Comments (if any, and for reference):

Reviewer's Responses to Questions

**Comments to the Author**

1. If the authors have adequately addressed your comments raised in a previous round of review and you feel that this manuscript is now acceptable for publication, you may indicate that here to bypass the “Comments to the Author” section, enter your conflict of interest statement in the “Confidential to Editor” section, and submit your "Accept" recommendation.

Reviewer #3: All comments have been addressed

2. Does this manuscript meet PLOS Global Public Health’s publication criteria? Is the manuscript technically sound, and do the data support the conclusions? The manuscript must describe methodologically and ethically rigorous research with conclusions that are appropriately drawn based on the data presented.

Reviewer #3: Yes

3. Has the statistical analysis been performed appropriately and rigorously?

Reviewer #3: Yes

4. Have the authors made all data underlying the findings in their manuscript fully available (please refer to the Data Availability Statement at the start of the manuscript PDF file)?

Reviewer #3: Yes

5. Is the manuscript presented in an intelligible fashion and written in standard English?

Reviewer #3: Yes

6. Review Comments to the Author

Reviewer #3: (No Response)

7. PLOS authors have the option to publish the peer review history of their article (what does this mean?). If published, this will include your full peer review and any attached files.

**Do you want your identity to be public for this peer review?** For information about this choice, including consent withdrawal, please see our Privacy Policy.

Reviewer #3: **Yes:** Rabia Hussain
